# A Simple yet Effective Baseline for Robust Deep Learning with Noisy Labels

## Abstract

Recently deep neural networks have shown their capacity to memorize training data, even with noisy labels, which hurts generalization performance. To mitigate this issue, we propose a simple but effective method that is robust to noisy labels, even with severe noise. Our objective involves a variance regularization term that implicitly penalizes the Jacobian norm of the neural network on the whole training set (including the noisy-labeled data), which encourages generalization and prevents overfitting to the corrupted labels. Experiments on noisy benchmarks demonstrate that our approach achieves state-of-the-art performance with a high tolerance to severe noise.

## 1 Introduction

Recently deep neural networks (DNNs) have achieved remarkable performance on many tasks, such as speech recognition Amodei et al. (2016), image classification He et al. (2016), object detection Ren et al. (2015). However, DNNs usually need a large-scale training dataset to generalize well. Such large-scale datasets can be collected by crowd-sourcing, web crawling and machine generation with a relative low price, but the labeling may contain errors. Recent studies Zhang et al. (2016); Arpit et al. (2017) reveal that mislabeled examples hurt generalization. Even worse, DNNs can memorize the training data with completely randomly-flipped labels, which indicates that DNNs are prone to overfit noisy training data. Therefore, it is crucial to develop algorithms robust to various amounts of label noise that still obtain good generalization.

To address the degraded generalization of training with noisy labels, one direct approach is to reweigh training examples Ren et al. (2018); Jiang et al. (2017); Han et al. (2018); Ma et al. (2018), which is related to curriculum learning. The general idea is to assign important weights to examples with a high chance of being correct. However, there are two major limitations of existing methods. First, imagine an ideal weighting mechanism. It will only focus on the selected clean examples. For those incorrectly labeled data samples, the weights should be near zero. If a dataset is under 80% noise corruption, an ideal weighting mechanism assigns nonzero weights to only 20% examples and abandons the information in a large amount of 80% examples. This leads to an insufficient usage of training data. Second, previous methods usually need some prior knowledge on the noise ratio or the availability of an additional clean unbiased validation dataset. But it is usually impractical to get this extra information in real applications. Another approach is correction-based, estimating the noisy corruption matrix and correcting the labels Patrini et al. (2017); Reed et al. (2014); Goldberger & Ben-Reuven (2017). But it is often difficult to estimate the underlying noise corruption matrix when the number of classes is large. Further, there may not be an underlying ground truth corruption process but an open set of noisy labels in the real world. Although many complex approaches Jiang et al. (2017); Ren et al. (2018); Han et al. (2018) have been proposed to deal with label noise, we find that a simple yet effective baseline can achieve surprisingly good performance compared to the strong competing methods.

In this paper, we first analyze the conditions for good generalization. A model with simpler hypothesis and smoother decision boundaries can generalize better. Then we propose a new algorithm which can satisfy the conditions and take advantage of the whole dataset including the noisy examples to improve the generalization.

Our main contributions are:

- We build a connection between the generalization of models trained with noisy labels and the smoothness of solutions, which is related to the subspace dimensionality.

- We propose a novel approach for training with noisy labels, which greatly mitigates over-fitting. Our method is simple yet effective and can be applied to any neural network architecture. Additional knowledge on the clean validation dataset is not required.

- A thorough empirical evaluation on various datasets (CIFAR-10, CIFAR-100) is conducted and demonstrates a significant improvement over the competing strong baselines.

## 2 PRELIMINARIES

In this section, we briefly introduce some notations and settings in learning with noisy labels.

The target is to learn a robust $K$-class classifier $f$ from a training dataset of images with noisy supervision. Let $\mathcal{D} = \{(x_1, \tilde{y}_1), ..., (x_N, \tilde{y}_N)\}$ denote a training dataset, where $x_n \in X$ is the $n$-th image in sample space $X$ (e.g., $\mathbb{R}^d$) with its corresponding noisy label $\tilde{y}_n \in \{1, 2, ..., K\}$.

### 2.1 LABEL NOISE

The label noise is often assumed to be class-conditional noise in previous work Natarajan et al. (2013); Patrini et al. (2017), where the label $y$ is flipped to $\tilde{y} \in \mathcal{Y}$ with some probability $p(\tilde{y}|y)$. It means that $p(\tilde{y}|x, y) = p(\tilde{y}|y)$, , the corruption of labels is independent of the input $x$. This kind of assumption is an abstract approximation to the real-world corruption process. For example, non-expert labelers may fail to distinguish some specific species. The probability $p(\tilde{y}|y)$ is represented by a noise transition matrix $T \in [0, 1]^{K \times K}$, where $T_{ij} = p(\tilde{y} = j|y = i)$. The examples $(x_i, \tilde{y}_i)$ in $\mathcal{D}$ are sampled from $p(x, \tilde{y}) = \sum_y p(\tilde{y}|y)p(y|x)p(x)$, a distribution that marginalizes over the unknown true label. A few exceptions Xiao et al. (2015); Menon et al. (2016) also consider the input-dependent noise model $p(\tilde{y}|x, y)$.

## 3 OUR APPROACH

In this section, we present a new robust training algorithm to deal with noisy labels. We argue that a model with lower complexity is more robust to label noise and generalizes well. The dimensionality of the learned subspace and the smoothness of decision boundaries can both indicate how complex the model is. Therefore, we propose a method to regularize the predictive variance to achieve low subspace dimensionality and smoothness, respectively.

### 3.1 VARIANCE-BASED REGULARIZATION

In order to alleviate over-fitting to the label noise, we propose a regularizer that is not dependent on the labels. We induce the smoothness of decision boundaries along the data manifold, which is shown to improve the generalization and robustness. If an example $x$ is incorrectly labeled with $\tilde{y}$, it has a high probability to lie near the decision boundary or in the wrong cluster not belonging to $y$. Therefore, the prediction variance can be high on the noisy examples. We propose to regularize the variance term. The mapping function is smoothed and thus also the decision boundaries.

Concretely, the variance is estimated by the difference of predictions under perturbations $\xi$ and $\xi'$ including the input noise like Gaussian noise and stochastic data augmentation, as well as the network noise like dropout:

$$R_V(\theta) = \frac{1}{N} \sum_{i=1}^{N} \mathbb{E}_{\xi', \xi} \|f(x_i; \theta', \xi') - f(x_i; \theta, \xi)\|^2. \tag{1}$$

We can show that $R_V(\theta)$ is an unbiased estimation of the predictive variance if the perturbations are treated as a part of the model uncertainty.

**Relation to the generalization of DNNs.** We show that this regularization helps to learn a low-dimensional feature space that captures the underlying data distribution. The variance term implicitly estimates the Jacobian norm, , the Frobenius norm of the Jacobian of the network output w.r.t. the

inputs: $\|J(x)\|_F$. A simplified version is to assume $\xi$ is sampled from a Gaussian distribution, i.e., $\xi, \xi' \sim \mathcal{N}(0, \sigma^2 I)$ and the perturbation is small and additive, , $\tilde{x} = x + \xi$ where $\sigma$ is near zero.

$$\lim_{\sigma \to 0} \frac{1}{\sigma^2} R_V(\theta) = \lim_{\sigma \to 0} \frac{1}{\sigma^2} \frac{1}{N} \sum_{i=1}^{N} \mathbb{E}_{\xi', \xi} \|f(x_i + \xi; \theta) - f(x_i + \xi'; \theta)\|^2 \quad (2)$$

By first-order Taylor expansion, and let $J(x) = \frac{\partial f}{\partial x}$

$$f(x + \xi) = f(x) + J(x)\xi + o(\xi), \quad (3)$$

and omitting the high-order terms, we have

$$\lim_{\sigma \to 0} \frac{1}{\sigma^2} R_V(\theta) = \frac{1}{N} \sum_{i=1}^{N} \text{Tr}(J(x_i)^\top J(x_i)) = \frac{1}{N} \sum_{i=1}^{N} \|J(x_i)\|_F^2.$$

If we further take expectation over $N$ samples of $x_i$, we get

$$\lim_{N \to \infty} \lim_{\sigma \to 0} \frac{1}{\sigma^2} R_V(\theta) = \mathbb{E}_x \|J(x)\|_F^2. \quad (4)$$

It can be proved that this is an unbiased estimator. For perturbations of natural images, similar analysis applies. It was shown in Sokolić et al.; Novak et al. (2018) that the Jacobian norm is related to the generalization performance both theoretically and empirically. Perturbations on the data manifold can be approximated by stochastic data augmentation. Similar objectives have been explored in semi-supervised learning Laine & Aila (2016); Tarvainen & Valpola (2017) but with different motivations.

**Relation to posterior regularization.** Minimizing the predictive variance has been applied to regression tasks Jean et al. (2018). It was pointed out that variance minimization can be explained in the framework of posterior regularization. Optimizing the objective is equivalent to computing a regularized posterior by solving a regularized Bayesian inference (RegBayes) optimization problem Zhu et al. (2014); Jean et al. (2018). It restricts the solution to be of some specific form, which is equivalent to imposing some prior knowledge of the model structure. The regularizer serves as an inductive bias on the structure of the feature space. By reducing the variance of predictions, the neural network is encouraged to learn a low-dimensional feature space where the training examples are far from the decision boundaries and tend to cluster together. This alleviates the possibility of the model to increase its complexity to fit the noisy labels.

Therefore, the learning objective is simply

$$\min_\theta \sum_{i=1}^{N} \ell(f(x_i; \theta), \tilde{y}_i) + \lambda R_V(\theta), \quad (5)$$

where the first term is any loss function including the cross-entropy loss or previously proposed noise-robust losses. In Section 4, we show empirically that the objective can learn a model with low subspace dimensionality and low hypothesis complexity.

## 4 EXPERIMENTS

In this section, we present both quantitative and qualitative results to demonstrate the effectiveness of our method. Our method is independent of both the architecture and the dataset.

### 4.1 EXPERIMENTAL SETUP

We first provide results on the widely adopted benchmarks, CIFAR-10 and CIFAR-100. Results on ImageNet and WebVision will be provided in Sec. 5.5 and Sec. 5.6. Following the settings in previous work Jiang et al. (2017); Ren et al. (2018), we train wide residual networks WRN-28-10 Zagoruyko & Komodakis (2016) for 200 epochs with mini-batch size 128. All the experiments are trained using momentum 0.9 and weight decay $1 \times 10^{-4}$. We use learning rate 0.1 and a cosine annealing schedule as suggested in Loshchilov & Hutter (2016). We use the implementation of WRN-28-10 in the official code of AutoAugment Cubuk et al. (2018).

Table 1: Averaged test error rates (%) and the standard deviations over 3 runs on CIFAR-10 under different uniform noise fraction. [†] marks methods trained using additional clean validation images. Best results are highlighted in bold.

| Methods | Noise Ratio $\eta$ | | | | | Network |
| | 0 | 0.2 | 0.4 | 0.6 | 0.8 | |
| --- | --- | --- | --- | --- | --- | --- |
| Bootstrap-hard Reed et al. (2014) | 10.94 ± 0.9 | 20.81 ± 0.4 | 23.33 ± 0.8 | 29.43 ± 0.3 | – | 12-layer CNN |
| Forward-correction Patrini et al. (2017) | 9.73 ± 0.0 | 15.39 ± 0.3 | 18.16 ± 0.1 | 27.59 ± 0.7 | – | 12-layer CNN |
| D2L Ma et al. (2018) | 10.59 ± 0.2 | 14.87 ± 0.6 | 16.64 ± 0.5 | 27.16 ± 0.6 | – | 12-layer CNN |
| Generalized Cross Entropy Zhang & Sabuncu (2018) | 6.5 | 10.13 ± 0.2 | 12.87 ± 0.22 | 17.46 ± 0.23 | 32.08 ± 0.6 | ResNet-34 |
| Co-teaching Han et al. (2018) | 6.05 | 17.68 | – | – | – | 13-layer CNN |
| MentorNet Jiang et al. (2017)[†] | 4 | 8 | 11 | – | 51 | WRN-101-10 |
| Learning to reweight Ren et al. (2018)[†] | 3.87 | – | 13.08 ± 0.19 | – | – | WRN-28-10 |
| Ours | 3.79 ± 0.13 | **3.87 ± 0.15** | **5.05 ± 0.24** | **6.42± 0.28** | **13.31 ± 0.45** | WRN-28-10 |

Table 2: Test error rates (%) on CIFAR-100 under different uniform noise fraction. [†] marks methods trained using additional clean images. Best results are highlighted in bold.

| Methods | Noise Ratio $\eta$ | | | | | Network |
| | 0 | 0.2 | 0.4 | 0.6 | 0.8 | |
| --- | --- | --- | --- | --- | --- | --- |
| Bootstrap-hard Reed et al. (2014) | 31.69 ± 0.2 | 41.51 ± 0.4 | 53.56 ± 0.7 | 57.35 ± 0.9 | – | ResNet-44 |
| Forward-correction Patrini et al. (2017) | 31.46 ± 0.1 | 39.75 ± 0.2 | 48.73 ± 0.3 | 55.78 ± 0.7 | – | ResNet-44 |
| D2L Ma et al. (2018) | 31.40 ± 0.3 | 37.80 ± 0.5 | 46.99 ± 0.7 | 54.79 ± 0.4 | – | ResNet-44 |
| Generalized Cross Entropy Zhang & Sabuncu (2018) | 28.6 | 33.19 ± 0.42 | 38.23 ± 0.24 | 45.96 ± 0.56 | 52.34 ± 0.69 | ResNet-34 |
| Co-teaching Han et al. (2018) | 29.15 | 45.77 | – | – | – | 13-layer CNN |
| MentorNet Jiang et al. (2017)[†] | 21 | 27 | 32 | – | 65 | WRN-101-10 |
| Learning to reweight Ren et al. (2018)[†] | 21.8 | – | 38.66 ± 2.06 | – | – | WRN-28-10 |
| Ours | 18.6±0.15 | **19.45 ±0.22** | **25.73 ± 0.47** | **38.23 ±0.52** | **44.68 ± 0.75** | WRN-28-10 |

## 4.2 Input-agnostic uniform label noise

First, we test on the uniform random label noise on CIFAR-10 and CIFAR-100. Following common practice Patrini et al. (2017); Jiang et al. (2017); Zhang & Sabuncu (2018), a certain percentage $\eta$ (0%, 20%, 40%, 60%, 80%) of true labels on the training dataset are replaced by random labels through uniform sampling. We report the averaged error rates on test datasets over 3 runs. Experimental results are summarized in Table 1 and 2. Note that different network architectures are used in the competing methods, as pointed out in the table. The error rates of the base networks in each method are shown in the second column of 0% noise (clean), where performance relative to the standard clean settings can be observed. We fix the hyper-parameter $\lambda = 300$ in all the experiments for CIFAR-10 and $\lambda = 3000$ for CIFAR-100.

In all the experiments, our method achieves significantly better resistance to label noise from moderate to severe levels. In particular, our approach attains a 13.31% error rate on CIFAR-10 with a noise fraction of 80%, down from the previous best 32.08%. Using the same network architecture WRN-28-10 as ours and 1000 clean validation images, learning to reweight Ren et al. (2018) achieves 38.66% test error on CIFAR-100 with 40% noise while our method achieves a better 25.73% even without any knowledge on the clean validation images. Figures 1 and 3 plot the test accuracy against the number of epochs on the two datasets. We provide a simple baseline – CCE, standing for categorical cross-entropy loss that treats all the noisy training examples as clean and trains a WRN-28-10. We can see that the baseline tends to over-fit the label noise at the later stage of training while our method does not suffer from the incorrect training signal.

## 5 Conclusion

We propose a simple but effective algorithm for robust deep learning with noisy labels. Our method builds upon a variance regularizer that prevents the model from overfitting to the corrupted labels. Extensive experiments given in the paper show that the generalization performance of DNNs trained with corrupted labels can be improved significantly using our method, which can serve as a strong baseline for deep learning with noisy labels.

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

## A    RELATED WORK

Learning with noisy labels has been broadly studied in previous work, both theoretically Natarajan et al. (2013) and empirically Reed et al. (2014); Han et al. (2018); Jiang et al. (2017). Here we focus on the recent progress on deep learning with noisy labels. Since DNNs have high capacity to fit the (noisy) data, it brings new challenges different from that in the traditional noisy label settings.

**Generalization of DNNs.** Previous works Zhang et al. (2016); Arpit et al. (2017); Shwartz-Ziv & Tishby (2017) find that DNNs have different learning patterns for clean or noisy labels. Zhang et al. (2016) shows that DNNs can easily memorize the training dataset even when the labels are random noise. An early stage of pattern learning and later memorization of noisy labels are observed in Arpit et al. (2017). Some previous work uses this kind of property to propose measures to modify the training process, such as the learned subspace dimensionality and the distribution of the loss values Ma et al. (2018); Han et al. (2018). Regularization techniques including dropout and early stopping have been shown to be effective to prevent over-fitting to noisy labels Arpit et al. (2017).

**Estimating noise distribution.** Many noisy estimation models have been proposed Natarajan et al. (2013); Xiao et al. (2015); Patrini et al. (2017); Vahdat (2017). Some works assume the true label is modeled by a latent variable while the noisy label is observed. EM-like methods have been proposed to alternate between the learning of noisy corruption process and the modeling. Backward and forward corrections Patrini et al. (2017) use an estimated noise transition matrix to modify the loss function. In general, the noise is assumed to be input-independent but class-dependent. Input-dependent noise has been explored in Menon et al. (2016); Xiao et al. (2015).

**Noise-robust loss functions.** The mean absolute error (MAE) was proposed as a noise-robust alternative to the cross-entropy loss Ghosh et al. (2017) but was known to be hard to converge. An extension and generalization of MAE, generalized cross entropy, was recently developed Zhang & Sabuncu (2018).

**Identifying clean examples.** Co-teaching Han et al. (2018) proposes to identify the examples with small loss as clean examples. Learning to reweight Ren et al. (2018) equals to shifting the training distribution $p(x, y)$ to match the clean validation distribution $q(x, y)$, that is to minimize $D_{f_\theta}(w(x, y)p(x, y), q(x, y))$ where $D$ is some distance measure implicitly learned by $f_\theta$ and $w(x, y)$ is the density ratio, the learned weights for each example $(x, y)$.

**Using additional clean validation dataset.** Azadi et al. (2015) proposed a regularization term to encourage the model to select reliable examples. Hendrycks et al. (2018) proposed Golden Loss Correction to use a set of trusted clean data to mitigate the effects of label noise. They estimate the corruption matrix using the trained network with noisy labels and then re-train the network corrected by the corruption matrix. Ren et al. (2018) also used a small clean validation dataset to determine the weights of training examples. The success of these methods is based on the assumption that clean data is from the same distribution as the corrupted data as well as the test data. However, more realistic scenario are ones where (1) $p(x)$ varies between the clean data and the noisy data, e.g., imbalanced datasets. 2) There is class mismatch: $p(y|x)$ differs. Similar problems exist in semi-supervised learning. All these methods require a clean validation dataset to work well while the proposed method does not require it.

We also plot label precision against number of epochs in Figure 4. Here we treat the $1 - \eta$ ratio of the training examples with minimal training losses as the clean examples. The label precision is computed as the portion of true clean examples among them. The ideal algorithm without over-fitting will have 100% label precision. The higher the label precision is, the better robustness the model achieves. Figure 4 demonstrates that our method obtains a higher label precision.

### A.1    CLASS-DEPENDENT ASYMMETRIC LABEL NOISE

A more realistic and more challenging noise type than the uniform noise is to corrupt between the semantically similar classes. For CIFAR-10, the class-dependent asymmetric noise is simulated by mapping TRUCK → AUTOMOBILE, BIRD → AIRPLANE, DEER → HORSE, CAT ↔ DOG, as done in Patrini et al. (2017); Zhang & Sabuncu (2018). The noise strength is controlled by the flipping

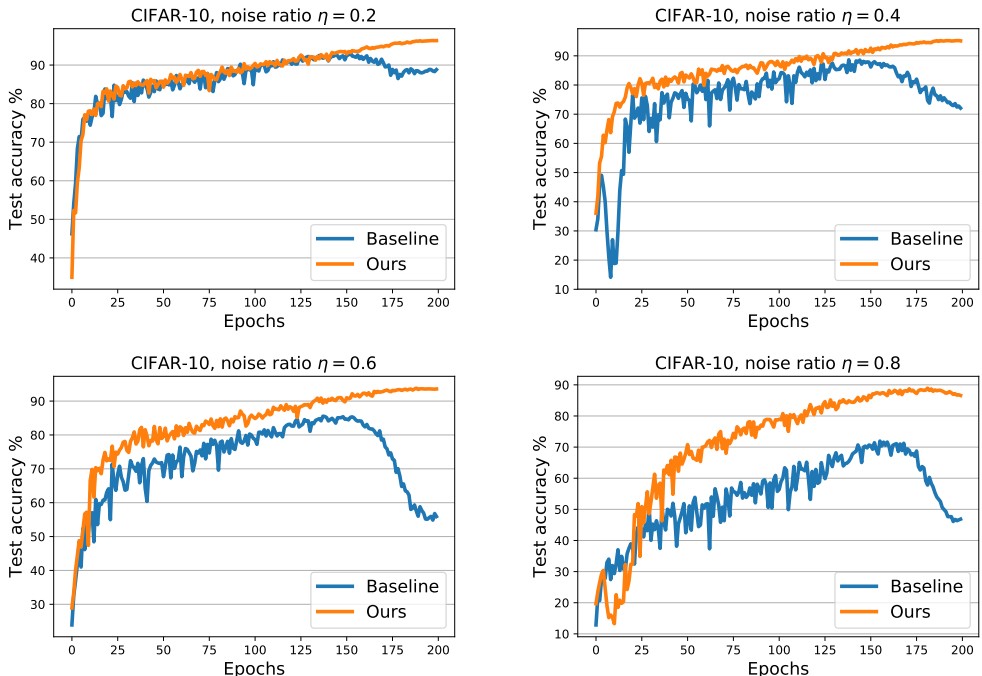

Figure 1: Test accuracy against the number of epochs on CIFAR-10 under different uniform noise ratio trained with WRN-20-10. Our method is less prone to the label noise over-fitting.

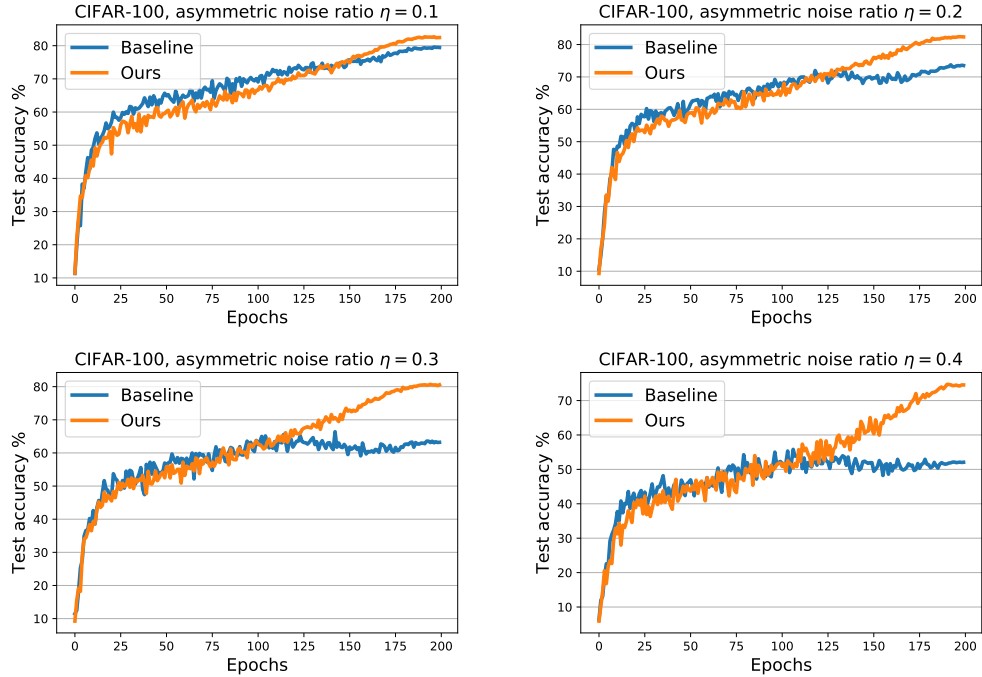

Figure 2: Test accuracy against the number of epochs on CIFAR-100 under different asymmetric noise ratios trained with WRN-20-10.

probability $\eta$. The noise transition matrix is:

$$
\begin{bmatrix}
1 & 0 & 0 & 0 & 0 & 0 & 0 & 0 & 0 & 0 \\
0 & 1 & 0 & 0 & 0 & 0 & 0 & 0 & 0 & 0 \\
\eta & 0 & 1-\eta & 0 & 0 & 0 & 0 & 0 & 0 & 0 \\
0 & 0 & 0 & 1-\eta & 0 & \eta & 0 & 0 & 0 & 0 \\
0 & 0 & 0 & 0 & 1-\eta & 0 & 0 & \eta & 0 & 0 \\
0 & 0 & 0 & \eta & 0 & 1-\eta & 0 & 0 & 0 & 0 \\
0 & 0 & 0 & 0 & 0 & 0 & 1 & 0 & 0 & 0 \\
0 & 0 & 0 & 0 & 0 & 0 & 0 & 1 & 0 & 0 \\
0 & 0 & 0 & 0 & 0 & 0 & 0 & 0 & 1 & 0 \\
0 & \eta & 0 & 0 & 0 & 0 & 0 & 0 & 0 & 1-\eta
\end{bmatrix}
\tag{6}
$$

Table 3: Results on CIFAR-10 and CIFAR-100 with class-dependent asymmetric noise. Averaged accuracy and standard deviation over 3 runs are reported. The results of competing methods are taken from Zhang & Sabuncu (2018). CCE stands for commonly-used categorical cross-entropy loss function, MAE stands for mean absolute error. Forward $T$ Patrini et al. (2017) uses the ground-truth noise transition matrix while Forward $\hat{T}$ Patrini et al. (2017) estimates $T$. Trunc $\mathcal{L}_q$ loss is a noise-robust loss function proposed in Zhang & Sabuncu (2018).

| Datasets | Methods | Noise Ratio $\eta$ | | | |
| --- | --- | --- | --- | --- | --- |
| | | 0.1 | 0.2 | 0.3 | 0.4 |
| CIFAR-10 | CCE | $90.69 \pm 0.17$ | $88.59 \pm 0.34$ | $86.14 \pm 0.40$ | $80.11 \pm 1.44$ |
| | MAE | $82.61 \pm 4.81$ | $52.93 \pm 3.60$ | $50.36 \pm 5.55$ | $45.52 \pm 0.13$ |
| | Forward $T$ Patrini et al. (2017) | $91.32 \pm 0.21$ | $90.35 \pm 0.26$ | $89.25 \pm 0.43$ | $\mathbf{88.12 \pm 0.32}$ |
| | Forward $\hat{T}$ Patriniet al. (2017) | $90.52 \pm 0.26$ | $89.09 \pm 0.47$ | $86.79 \pm 0.36$ | $83.55 \pm 0.58$ |
| | Trunc $\mathcal{L}_q$ Zhang & Sabuncu (2018) | $90.43 \pm 0.25$ | $89.45 \pm 0.29$ | $87.10 \pm 0.22$ | $82.28 \pm 0.67$ |
| | Baseline (CCE) | $94.31 \pm 0.19$ | $90.29 \pm 0.35$ | $84.61 \pm 0.41$ | $78.24 \pm 0.82$ |
| | Ours | $\mathbf{95.69 \pm 0.18}$ | $\mathbf{94.01 \pm 0.22}$ | $\mathbf{92.44 \pm 0.37}$ | $85.62 \pm 0.77$ |
| CIFAR-100 | CCE | $66.54 \pm 0.42$ | $59.20 \pm 0.18$ | $51.40 \pm 0.16$ | $42.74 \pm 0.61$ |
| | MAE | $13.38 \pm 1.84$ | $11.50 \pm 1.16$ | $8.91 \pm 0.89$ | $8.20 \pm 1.04$ |
| | Forward $T$ Patrini et al. (2017) | $71.05 \pm 0.30$ | $71.08 \pm 0.22$ | $70.76 \pm 0.26$ | $70.82 \pm 0.45$ |
| | Forward $\hat{T}$ Patrini et al. (2017) | $45.96 \pm 1.21$ | $42.46 \pm 2.16$ | $38.13 \pm 2.97$ | $34.44 \pm 1.93$ |
| | Trunc $\mathcal{L}_q$ Zhang & Sabuncu (2018) | $68.86 \pm 0.14$ | $66.59 \pm 0.23$ | $61.87 \pm 0.39$ | $47.66 \pm 0.69$ |
| | Baseline (CCE) | $79.40 \pm 0.22$ | $73.50 \pm 0.21$ | $63.02 \pm 0.32$ | $52.06 \pm 0.71$ |
| | Ours | $\mathbf{82.55 \pm 0.24}$ | $\mathbf{82.34 \pm 0.20}$ | $\mathbf{80.55 \pm 0.26}$ | $\mathbf{74.54 \pm 0.64}$ |

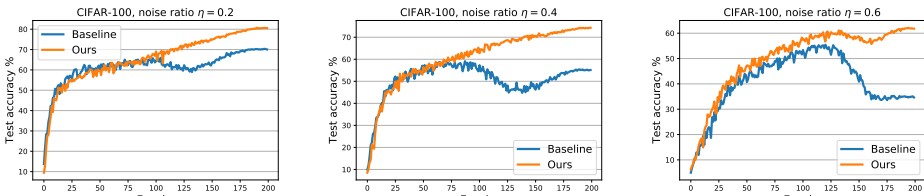

Figure 3: Test accuracy against the number of epochs on CIFAR-100 under different uniform noise ratios trained with WRN-20-10. Our method is less prone to label noise over-fitting.

For CIFAR-100, class dependent noise is simulated by flipping each class into the next class with probability $\eta$. The last class is flipped to the first class circularly, , the transition matrix has $1 - \eta$ on the diagonal and $\eta$ off the diagonal:

$$
\begin{bmatrix}
1-\eta & \eta & 0 & 0 & \cdots & 0 \\
0 & 1-\eta & \eta & 0 & \cdots & 0 \\
0 & 0 & 1-\eta & \eta & \cdots & 0 \\
\vdots & \vdots & \vdots & \ddots & \vdots & \vdots \\
0 & 0 & 0 & 0 & 1-\eta & \eta \\
\eta & 0 & 0 & 0 & 0 & 1-\eta
\end{bmatrix}
\tag{7}
$$

Results are presented in Table 3. We compare to a range of competing loss-correction methods whose results are taken from Zhang & Sabuncu (2018) and our baseline trained with only CCE. We use the same hyper-parameter $\lambda = 300$ among all the experiments for CIFAR-10 and $\lambda = 3000$ for CIFAR-100. Note that Forward $T$ is the forward correction Patrini et al. (2017) using the ground-truth noise transition matrix, whose results are almost perfect. Our method does not use any ground-truth knowledge of the noise corruption process. We can see that our method is robust to all the settings and is less influenced by the variations of noise types. The test accuracy along the training process on CIFAR-100 is also plotted in Figure 2.

## A.2 HYPER-PARAMETER SENSITIVITY ANALYSIS

We assess the sensitivity of our algorithm with respect to the hyper-parameter $\lambda$ and the results are plotted in Figure 5. We can see that the performance of our method remains stable across a wide range of hyper-parameter choices.

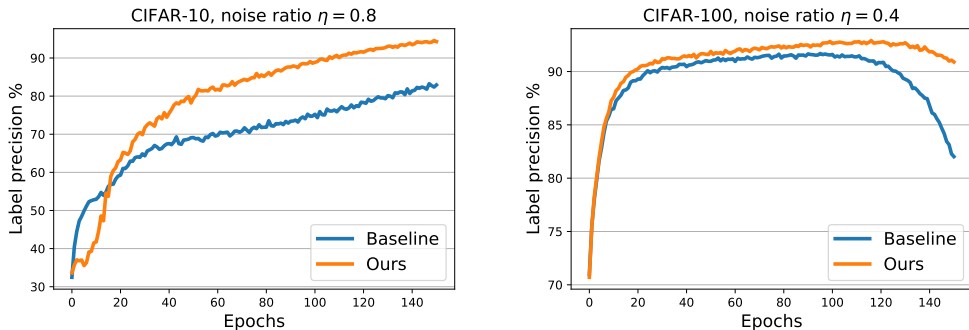

Figure 4: Label precision against the number of epochs on CIFAR-10 (left) and CIFAR-100 (right) with uniform noise, respectively. Here the label precision is computed by the percentage of clean training examples within those having $1 - \eta$ minimal training losses.

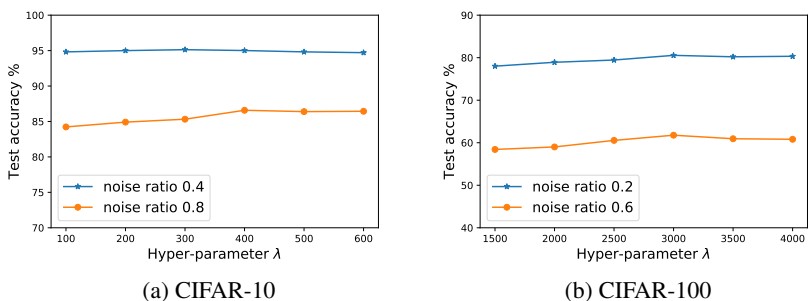

Figure 5: Hyper-parameter sensitivity analysis on CIFAR-10 and CIFAR-100 with uniform label noise. Our method is insensitive to a wide range of values for $\lambda$.

## A.3 VISUALIZATION

We visualize the embeddings of our algorithm on test data. Figure 6 shows the representations $h(x) \in \mathbb{R}^{128}$ projected to 2 dimension using t-SNE Maaten & Hinton (2008).

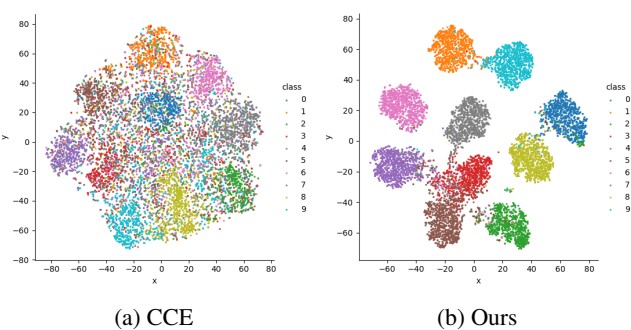

(a) CCE                                   (b) Ours

Figure 6: t-SNE 2D embeddings of the test dataset on CIFAR-10 trained with 60% uniform label noise. Each color represents a class. Our method learns a more separable feature space.

