# OpenReview forum: "A Simple yet Effective Baseline for Robust Deep Learning with Noisy Labels"
_ICLR.cc/2019/Workshop/LLD — Submitted to LLD 2019_

### Official Review · AnonReviewer2 · 2019-04-06
**Interesting method with excitingly good empirical results;  Paper presentation can be improved further**

**Rating:** 3
**Confidence:** 1

**Review:**

This paper presents a method to train deep NN with noisy labels by adding a variance regularization term.  The authors show/derive that such variance regularization term is an unbiased estimator of Jacobian norm of the neural network. As previous literature (Sokolic et al; Novak et al. 2018) show that this Jacobian norm is highly related to the generalization performance of NN, the authors conclude that minimizing their proposed variance regularization term can also help model generalization.  Finally, the authors conduct experiments on CIFAR-10 AND CIFAR-100 datasets and show their proposed learning objective can be used to train a NN robust to noise. The experimental results are strikingly good.

Overall the paper is interesting, the major question is why EXACTLY minimizing the norm of Jacobian of NN can help improve the model robustness. I am not very familiar with the work (Sokolic et al; Novak et al. 2018) and thus I would appreciate a (high-level) description of those previous studies' major ideas. Furthermore, I am wondering why the authors treat good "generalization" of NN as the same as good "robustness" of NN. The authors describe label noise in Section 2.1 but I don't see how such label noise is modeled later in the regularization term design. Finally, I am a little bit confused about the derivation of the equation between equation (3) and (4), from lim_{\tau -> 0} ... to \frac{1}{N} \sum_{i=1}^{N} Tr(...). More explanation on this part is appreciated.

The structure of the paper is good and writing is overall clear. There are still some places can be improved. First, at the begining of section 2, the authors write a K-class classifier f from ..., it's better to explicitly state the function f is from R^{d} to R^{K}, instead of being a scalar-valued function. This can make later discussion on Jacobian matrix more clear. Second, the connection of section 2.1 with later parts is not clear. Finally, there are some typos listed below:
1. in the third line of section 2.1, there are two contiguous ","s .
2. in the last line of page 2, there are two contiguous ","s .
3. in the second line of page 3, there are two contiguous ","s .
4. in the second line of section 4.1, what are the "Sec. 5.5 and Sec. 5.6"?

---

### Official Review · AnonReviewer1 · 2019-04-09
**Impressive performance but contributions unclear**

**Rating:** 2
**Confidence:** 2

**Review:**

The authors propose a method for learning deep neural networks under label noise by regularizing the Jacobian of the network as a first-order approximation of the variance between perturbed examples. They argue that this will mitigate overfitting to mislabeled examples and produce smoother decision boundaries. They evaluate their methods by comparing to other noise-robust techniques on CIFAR-10 and CIFAR-100.

The authors present a complete piece of work, from a summary of noise issues in deep learning, to a description of their variance regularization technique, to their experiments. The paper follows a logical structure, and is generally well written. It would benefit from a clearer discussion of related works in the main body.

The regularization term they use is sensible, and produces strong results on the datasets in comparison to the selected methods. The results are continued in the Appendix in the data-dependent noise case. Had the authors focused on empirical variance regularization results for image classification, this would have been a much stronger paper.

However, the authors don't deliver on their two main stated contributions. The first contribution - analysis of solution smoothness and subspace dimensionality for noisily trained models - is not presented. They reference past works (e.g. Sokolic et al., 2016) but do not provide an original analysis or evaluation as implied ("we show empirically that the objective can learn a model with low subspace dimensionality and low hypothesis complexity"); their experiments solely measure classification performance. The authors' second contribution is to "propose a novel approach for training with noisy labels. ". However, variance regularization via data augmentation has been previously proposed ("Regularization With Stochastic Transformations and Perturbations for Deep Semi-Supervised Learning", Sajjadi et al., 2016), as has Jacobian regularization ("Robust Large Margin Deep Neural Networks", Sokolic et al., 2016). Jacobian regularization and related methods have also been previously analyzed in the noisy label setting ("Gradient Regularization Improves Accuracy of Discriminative Models", Varga et al., 2017). If the authors' proposed method is significantly different from these approaches, it is not clear from the paper.

Again, had this paper focused on empirical evaluation of Jacobian regularization for deep image recognition as compared to other methods, it would have been stronger. In its current form, the contributions are unclear.

---

### Decision · Program_Chairs · 2019-04-16
**Acceptance Decision**

**Decision:**

Reject

**Comment:**

Reviewers found issue with the novelty, further clarification of novelty is needed